# On the Influence of Linear Energy/Heat Input Coefficient on Hardness and Weld Bead Geometry in Chromium-Rich Stringer GMAW Coatings

**DOI:** 10.3390/ma15176019

**Published:** 2022-08-31

**Authors:** Jan Pawlik, Jacek Cieślik, Michał Bembenek, Tomasz Góral, Sarken Kapayeva, Madina Kapkenova

**Affiliations:** 1Faculty of Mechanical Engineering and Robotics, AGH University of Science and Technology, A. Mickiewicza 30, 30-059 Krakow, Poland; 2School of Mechanical Engineering, D. Serikbayev East Kazakhstan Technical University, Ulitsa Serikbayeva 19, Ust-Kamenogorsk 070000, Kazakhstan

**Keywords:** hardfacing, GMAW, WAAM, C45 steel, heat input, chromium based cored wires

## Abstract

Wear of the working surfaces of machinery parts is a phenomenon that cannot be fully countered, only postponed. Among surface lifecycle elongation techniques, hardfacing is one which is most often used in heavy load applications. Hardfaced coating can be applied using different welding approaches or thermal spraying technologies, which differ when it comes to weld bead dimensional precision, layer thickness, process efficiency and material. In this study the authors examine the geometrical behavior and hardness properties of two distinctive chromium-based Gas Metal Arc Welding (GMAW) cored wires. The stringer beads are applied numerically with five levels of linear energy, being a resultant of typical values of welding speed and wire feed, ranging between 250 mm/s to 1250 mm/s (welding speed) and 2 m/min to 10 m/min (wire feed). The samples were cut, etched and measured using a digital microscope and Vickers indenter, additionally the chemical composition was also examined. Hardness was measured at five points in each cutout, giving 40 measurements per sample. The values were analyzed using an ANOVA test as a statistical background in order to emphasize the divergent behavior of the cored wires. It appeared that, despite having less chromium in its chemical composition, wire DO*351 exhibits higher hardness values; however, DO*332 tends to have a more stable geometry across all of the heat input levels.

## 1. Introduction

The advancement in almost any branch of the industry is inevitably impaired by futile fight against machine wear, especially in the heavy industry, such as mineral extraction and processing. The estimation of the annual cost of wear worldwide can be roughly assessed to be billions of dollars [1]. The degradation of machine parts is a phenomenon that usually starts on superficial layer of a given element [2,3]. This process deteriorates the part geometry, which leads to further mass loss or volumetric deviations, eventually resulting in machine failure. The wear can be caused by a wide range of causes, mainly due to the abrasion [4], erosion, adhesion, fatigue, tribochemical reaction or combination of the aforementioned factors [5].

In many cases, the wear can be countered via hardfacing processes during machine maintenance. Hardfacing is a process, which uses heat to melt metallic (or ceramic-metallic) material and diffuse it with the base metal [6]. Initially it was used to repair iron casts; however, the development of the material science allowed for coating the base material with padding with distinctively different physical properties [7]. Depending on the thickness of the padding, one can utilize either welding [8] or thermal spraying technologies [9,10]. In both cases, the proper filler material has to be chosen to best suit the particular needs; sometimes the regenerated surface has to be soft (e.g., bronze) and in other occurrence it needs to be hard enough to withstand the abrasive environment.

When high hardness is required, one of the best choices for regeneration might be a cored wire, used in GMAW (Gas Metal Arc Welding) technology due to its availability, cost-effectiveness and relative easiness of application. The resulting hard coating can have higher resistance to abrasion than the base material (usually a construction steel), thus the part, being a quasi-composite, can achieve the desired properties at low effort [11,12].

In contrast to the welding metal components, in hardfacing, low dilution is required. High dilution means that the physical properties of the coating can be observably affected by the properties of the base material [13]. In order to achieve assumed geometrical features, finding the proper technological parameters is critical. Other research teams [12] took into account the influence of the current, voltage, welding speed, wire feed and the welding gun orientation on the quality of the resulting padding; however, optimal parameters have to be found both for the particular part application and every pair of base and filler materials. The mathematical model for predicting the weld bead shape, developed by Kumari and Singh [14], matches the experimental results nearly perfectly; however, their studies were focused only on 140 MXC welding wire, thus their model is not universal. The studies of Nagesh and Datta [15] and Siva et al. [16] used the neural network and genetic algorithms to predict the geometry, yet their studies did not focus on the conservation of hardfaced material volume.

One of the possible approaches to obtain evenly applied GMAW coating is to use a technique called WAAM—Wire Arc Additive Manufacturing [17]. It relies on adding consecutively applied weld beads, where the welding gun position is controlled numerically [18]. It allows for a precise control of the welding speed [19], manually achieved only by experienced welders. Comparing to other metal-coating techniques, WAAM coating application is efficient, since depending on the process circumstances it can reach from 1 kg up to 4 kg per hour, preserving 2 mm geometrical tolerancing and minimizing the welding defects [20]. However, when precise geometry is required, either additional machining (e.g., milling, grinding) might be necessary [21] or one has to carefully pick the technological parameters to achieve desired weld bead shape [22]. Examples of WAAM-fabricated products are presented on Figure 1. Subfigure (a) has raw external surface, whilst subfigure (b) depicts a part with externally machined walls.

Despite the huge advantages of that technology, one has to take into account several additional factors, mainly connected with heat input and heat dissipation [23,24]. Parameters such as voltage, free stick-out and type and flow of the inert or active gas are responsible mainly for the cross-sectional features. An incorrect set of parameters may lead to a variety of welding defects, including cracking, undercutting [25], lack of fusion [26,27] or loss of desired physical properties [27,28]. 

Moreover, geometrical deviation of the final part consists of the sum of deviations of a singular stringer bead and the deviations of solidifying weld beads layered onto excessively heated base of previous layers [29,30]. This problem is partially solved by the application of Fronius Cold Metal Transfer technology, nevertheless that technology is patented, expensive and excludes other current sources [31,32,33,34]. Another approach is to introduce an additional cooling subsystem [35], yet that can result in unwanted changes in the microstructure [36,37], although Reisgen et al. reported that they observed no negative effects [38].

The key factor in hardfacing process efficiency is the possibility to apply the highest amount of filler material in the shortest time possible, preserving the tribomechanical qualities of the coating [39]. In order to properly model the resulting weld bead geometry, a series of experiments for a particular material group is required. The variable input parameters, set by the operator, result in certain level of energy, delivered to the given material volume at a given time. In the welding industry, this resultant parameter is called either linear energy (also “arc energy”) or heat input. The linear energy parameter is a more theoretical variable, since it describes the output energy of the power source, whereas the heat input is the linear energy modified by the welding method thermal efficiency coefficient η. In the case of Submerged Metal Arc Welding (SMAW), this coefficient is equal to 1, whilst every other method suffers from some heat loss. The least efficient methods are Tungsten Inert Gas (TIG) and Plasma Arc Welding, having the η parameter equal to 0.6 [40]. This method has η = 0.8, making it a method of high efficiency, portability (on the contrary to SMAW), versatility. The fact that the filler material comes in a long form and is uniform and in a diameter spooled wire makes it also relatively easy to robotize.

In this paper, the authors focus on the geometrical properties and hardness of chromium-based cored wires utilized in GMAW welding approach. This kind of materials can be used in coating of machine parts, which are subjected to an abrasive working environment, such as drilling heads or tangential rotary picks in the mining industry [41,42], as seen in the example presented in Figure 2. The process of coating such mining tools relies on the combination of a conventional GMAW technique and the WAAM approach, since it requires relatively narrow geometrical tolerances resulting from numerically controlled indexed contour hardfacing [43].

In the current study, the C45 steel samples were coated with singular stringer weld beads, applied with a variable heat input parameter. The heat input parameter was a result of adjusting the welding speed and wire feed. The samples were cut into slices, etched and measured via a digital microscope. Additionally, in order to estimate the utility features, Vickers hardness was measured in five points of every cross-sectional cutout. This study is designed to provide the information about the influence of the technological parameters on the resultant weld bead physical properties. The geometrical constrain assumptions present in Section 2.5 were made according to the geometry of the mining conical pick, intended to use in the future, wear-related study.

## 2. Materials and Methods

### 2.1. Setup and Specimen

In this study, the authors examined two separate material types of chromium-based hardfaced layers applied on C45 (DIN 1.0503/ISO 683-1 1987) steel bars. The chemical composition of this base material, provided by the manufacturer is presented in Table 1 [44].

The carbon equivalent value for this type of material sits between 0.52–0.82, depending on the chosen CE calculation standard. This means, that the steel presents fair to poor weldability; however, the CE coefficient of 0.82 is calculated for the worst-case scenario.

All of the bars were cut and grinded to 115.0 × 34.5 × 23.5 mm and had singular, 90 mm long stringer bead applied with various volumetric material feed level. All of the samples were cooled in the free-air. In order to inspect the physical properties of beads, the Vicker’s hardness test and microscopic measurements of the cross-sectional geometry were performed. Figure 3 depicts the method of bead application and manner of cut-out preparation. The samples were cut in 7 places on a Struers Labo-Tom 5 metallographic cutting machine (Struers, Copenhagen, Denmark), resulting in eight slices per sample of available cross-section area.

The angle between the gun axis and the material surface was set to 90°. The hardfacing process was conducted via an original numerically controlled machine, designed and build by Jan Pawlik (the author) and located at AGH University of Science and Technology (Figure 4). It allowed the researchers to precisely control the wire feed and the velocity of the GMAW welding gun. The three-axis machine (with optional fourth, rotational axis) device offers 320 × 300 × 170 mm work area and head speed control from 0.1 mm/s to 200 mm/s. The CAD model and photograph of the machine during the process is available below.

The CNC Hardfacing Machine is a cartesian robot with a GMAW welding gun, attached to the Kemppi X8 MIG welding machine (Kemppi, Lahti, Finland). The process relies on Duet2Wifi control (Duet3D, Peterborough, UK) board with modified software. The modifications of the software are minor, and they are limited to adding a special G-code command to ignite and supress the electric arc. Yet, this experiment can be re-run using any linear actuator with speed control.

### 2.2. Base Material

In this experiment, the base material selected for hardfacing is the C45 medium carbon steel, also known as AISI 1045 or 1.0503 steel.

### 2.3. Filler Metal Material

As for the filler metal, the authors selected two different metal-cored welding wires, manufactured by Castolin Eutectic, namely ENDOTEC DO*351 and ENDOTEC DO*332 The authors will, from now on, refer to them as Wire A (DO*351) and Wire B (DO*332), respectively. Both are rich in chromium and are applicable to an abrasive working environment. The complete chemical compositions of the welding wires, provided by the manufacturer, are presented in the Table 2.

Both of the wires are 1.6 mm in diameter; however, as they are not solid-cored wires, their actual cross-sectional is slightly smaller and may vary a little across the length of the spool. This is important, considering further calculations.

### 2.4. Chemical Composition Analysis

The study of the weld beads was carried out on a JSM-6390LV scanning electron microscope manufactured by JEOL Ltd. (Tokyo, Japan) with INCA Energy Penta FET X3 energy-dispersive microanalysis system from OXFORD Instruments Analytical Limited (Abingdon, Oxfordshire, UK). The data collection points were picked randomly in three distinctive points on the area of weld bead cross-section several times and averaged.

### 2.5. Hardfacing Process Parameter Assumptions

The general geometry of the bead by its volume is dependent mainly on three factors: wire diameter, wire feed and welding gun speed. Having chosen a particular value of wire diameter and the desired output bead dimensions, only two factors are left to assess. In this experiment, the authors wanted to study the influence of the volumetric material flow rate on the physical properties of the hardfaced coatings.

Firstly, the input and output variables were translated into following equation: (1)$$Vol_{w}-Vol_{a}=Vol_{b}-Vol_{s}\;$$
where: 

*Vol_w_*—volume of the wire,

*Vol_a_*—volume of free space locked in the cored wire among the powder,

*Vol_b_*—volume of the formed stringer bead,

*Vol_s_*—volume of the spatter.

The volume of the atmosphere in the cored wire is hard to estimate, since it can be subtracted from stochastically occurring spatter. Another important factor to keep track of is the size and shape of the ending weld puddle.

The cross-sectional area of the welding wire can be expressed as follows:(2)$$A_{w}=\frac{\pi *D_{w}{}^{2}}{4}$$
where:

*A_w_*—area of the wire cross-section [mm^2^],

*D_w_*—diameter of the wire (here: 1.6 mm) [mm],

and the assumed cross-sectional area of the weld bead can be expressed as half of the cross-section of an ellipse:(3)$$A_{b}=\frac{\frac{\pi *h*w}{2}}{2}=\frac{\pi *h*w}{4}$$
where:

*A_b_*—cross-sectional area of the bead [mm^2^],

*h*—height of the bead (here: 3.2 mm) [mm],

*w*—width of the bead (here: 6.4 mm) [mm],

Since the time needed for “extruding” the wire and making the bead is the same, following formulas can be developed:(4)$$\begin{array}{c} Vol_{w}=A_{w}*v_{w}*t\\ Vol_{b}=A_{b}*v_{b}*t\\ A_{w}*v_{w}*t=A_{b}*v_{b}*t\;\;\;\;\;|:t\\ A_{w}*v_{w}=A_{b}*v_{b}\\ \frac{A_{w}}{A_{b}}=\frac{v_{b}}{v_{w}} \end{array}$$
where:

*Vol_w_*—volume of the extruded wire [mm^3^],

*Vol_b_*—volume of the applied coating [mm^3^],

*v_b_*_—_velocity of welding head [mm/min],

*v_w_*—wire feed [mm/min],

The volume of wire and volume of the bead can be further represented as the product of cross-sectional area and the distance. In turn, the distance can be assumed to be the product of time of bead application and the velocity value, thus finally, one can obtain the following simplified equation:(5)$$\frac{D_{w}{}^{2}}{h*w}=\frac{v_{b}}{v_{w}}=R_{vel}$$
where:

*R_vel_*—velocity ratio parameter.

In the current study, the authors wanted to achieve a relatively small and semi-circular weld bead; therefore, it was assumed, that the width will be double the height, equal to 3.20 mm. Consequently, the ratio between the wire feed and welding head velocity should be equal to 1:8. Considering the practical possibilities of the Kemppi power source, the authors decided to set the *R_vel_* parameter range according to the data in Table 3. The other parameters, kept as constant during every run, are assembled in Table 4.

The heat input factor, which is linear energy coefficient modified by the welding method efficiency factor, was calculated using the formula below [47]:$$Q=\eta *\frac{I*U}{v_{b}*1000}$$
where:

*Q*—heat input [kJ/mm],

*I*—current [A],

*U*—voltage [V],

*V_b_*—welding speed [mm/min],

*η*—welding method efficiency factor (in the case of GMAW it is equal to 0.8).

It should be noted that, according to the formula above, the heat input will be inversely proportional to the set of wire feed and welding speed, as the product of increasing the voltage and current coefficients will be smaller than the product of even a slight increase in speed multiplied by one thousand. Thus, the lowest values of velocities represent the highest heat input, and the highest values of velocities correspond to lowest energy introduced into the base material.

### 2.6. Geometric Measurements of the Weld Bead Dimensions

The measurements of the applied coatings geometry were carried out in the following manner: each steel bar was cut into eight pieces; however, the first one was cut 20 mm from the one end of the bar, and the last was cut 50 mm from the other end. This resulted in obtaining geometry measurements from the most important weld bead sections, excluding the bead start and end; the other slices were cut in 8 mm intervals.

The measured dimensional features of the cross-sections were the following: bead width, bead height, depth of penetration, area of the penetration and area of the padding (Figure 5). An example of the optical measurement of the geometry performed on Keyence VHX-7000 digital microscope (Keyence, Osaka, Japan) is provided below.

### 2.7. Hardness Measurements

The hardness test was performed on a Struers DuraScan-4 (Struers, Copenhagen, Denmark) via the Vickers HV5 method. Every cut-out was examined in five points, whereas the level of base material was set as the initial reference, from which +1, +2, −1 and −2 distance of indentation point were measured (Figure 6). Additionally, another measurement was performed at −3 mm, in order to estimate the average hardness of the whole material—in the current setup, the penetration of the coating was not deep enough to reach that level.

### 2.8. Statistical Background

The results were subjected to a one-way ANOVA test in Microsoft Excel 2020 with a Data Analysis Toolpack, which helped to establish if the volumetric flow rate (or linear energy coefficient) does or does not have an influence on the bead geometry and hardness for each of the hardfaced materials. The confidence level in this case was set to α = 0.05.

## 3. Results

### 3.1. Chemical Composition Analysis

The outcome of the chemical analysis of the arc-melted coatings for both of the materials is presented in Table 5. The device was not able to detect the lightweight elements, such as carbon; nevertheless, the composition of the heavier elements matches the datasheet provided by the manufacturer.

### 3.2. Weld Bead Cross-Sectional Geometry Measurements

As stated in Section 2.6, the dimensional measurements were performed on eight cut-outs for every steel bar. The quantitative results are aggregated for Wire A in the Appendix A (Table A1, Table A2, Table A3, Table A4 and Table A5 for L1A, L2A, L3A, L4A and L5A), Wire B in the Appendix B Table A6, Table A7, Table A8, Table A9 and Table A10 for L1B, L2B, L3B, L4B and L5B). The graphical overview of the microscopic cross-sectional images is presented on the Figure 7 (Wire A) and Figure 8 (Wire B). Those figures are enclosed to help visualize the different behaviors of the materials manufactured with different heat input levels. The number in columns (1–8) refer to the sample-cutting scheme, shown previously on Figure 3. The L1A row represents the run with incomplete fusion, meaning that the heat input was too low to enable the base metal to melt and mix with the coating.

The quantitative data of the dimensional measurements is presented on Figure 9 and Figure 10, where the values were averaged.

The dimensional measurements of the cross-sections of the beads show that, despite the same input parameters, the two inspected materials behave thoroughly different. Wire A manifests low, but with observable variability, in the case of every studied dimension; additionally, a “sweet-spot” can be indicated at around 8000 mm/min. wire feed and 1000 mm/min. welding speed.

Wire B exhibits inherently indifferent geometry, regardless of the material feed and welding speed. Despite recognizable variation in the heat input, a solidifying weld pool forms in a stable manner. Yet, at every measured point, the cross-sectional area of any Wire B bead is lower than the assumed value—it is connected with observable amount of spatter in this material. A visual representation can be seen in Figure 11.

### 3.3. Hardness Evaluation

The Vickers hardness, measured for both wires according to Figure 6 is presented in the form of a grayscale heatmap in Figure 12. The “avg” values stand for “horizontal mean hardness”. Considering the information provided in Section 3.2, the A1 row of Wire A shows correct data, as the hardfaced material did not melt and dilute with the base material properly. In general, even though Wire B contains more chromium and carbon in its chemical composition, it appears to be softer than Wire A. The hardness of the base material was measured in every sample approx. 1 mm below the border of Heat Affected Zone, and was assessed to be about 200–210 HV.

The detailed heatmap of hardness provided the authors with enough information to estimate the most suitable parameters for the Wire A, whereas the behavior of Wire B exhibits stable hardness (but lower than Wire A) across whole input parameter set. A simplified, averaged data is presented in Figure 13. The light grey squares in the A1 dataset correspond to the observed lack of proper fusion, which is considered as a welding defect.

### 3.4. Statistical Background

The measurements of the bead geometry and hardness have been subjected by a one-way ANOVA test, in order to estimate the significance of differences between output values. Firstly, the authors tested the differences between cross-sectional geometry, with respect to the heat input level for each wire. Secondly, the hardness levels were tested; however, please note that sample A1 was removed from ANOVA due to lack of penetration, thus being an outlier. The results of consecutive the ANOVA tests are presented in the Table 6.

## 4. Discussion

The two wires used in this study exhibit different reactions to the change in the heat input level. In the inspected scope of the input parameters, wire A tends to achieve higher hardness (with average value at approx. 780 HV5), while wire B not only achieves lower hardness (average at approx. 600 HV5), but also its hardness tends to decrease in an inversely proportional manner to the value of arc energy. That material property is coherent with the study of Das et al. [48], where the research team claimed that higher temperatures deteriorate the resultant hardness, severely. In the study of Savic and Cabrillo [49], the higher heat input also affects the coating hardness, although their filler material’s chemical composition was different. This phenomenon, however, is not observed in the wire A, which contains three times less chromium in its composition.

Wire B, however, presents an interesting dimensional stability across all heat input levels. Its cross-section shape always retains an elliptical shape, whereas Wire A’s cross-sectional shape varies from half-circular shape to a bell curve-type shape. The authors suspect that it might be related to the amount of chromium in the respective alloys, given that chromium requires higher temperature to melt. Technically, to achieve flatter geometry of Wire B bead, one should increase the arc energy; yet, as presented in Figure 8 L5B, the energy above 0.514 kJ/mm results in some gas inclusion—to avoid that one should maintain the heat input below 0.5 kJ/mm and adjust the voltage.

Although Kaewsakul et al. [50] and Ibrahim et al. [51] use different filler metal and they do not use the term “heat input” or “linear energy”, the key dependencies of bead physical properties on the GMAW technological parameters are similar to the conclusions drawn by the authors of the current study. The value of depth of penetration rises along with the length of the weld bead in both wires and stabilizes after 30–40 mm of bead length, which was expected. The depth is also proportional to the heat input level, in the case of both wires with the exception of L1A, where the energy of 0.2 kJ/mm was too low to melt and fuse the two materials. This value sets the lower boundary for the technological parameters, when using this material. It is noteworthy that the authors agree with Henckell et al. [52] that reduction in the energy input is advantageous when dealing with WAAM technology, thus the authors recommend targeting the lowest possible energy resulting in high quality beads—in the case of wire A, it appears to be set at around 0.3 kJ/mm.

Despite that in both cases, the width of the stringer bead did not met the requirements (besides L2A), the ANOVA analysis shown that the Wire B (F_calc_ = 11.05, F_crit_ = 2.64) is generally more reliable than the Wire A (F_calc_ = 204.73, F_crit_ = 2.64). Considering the assumed height, Wire B also has a higher score of compliance and, additionally, the means are statistically similar. This is the singular null hypothesis, which was eligible to accept, while others were to be rejected. The measured heights of Wire A beads are significantly different.

The second part of the current study, namely, the hardness inspection, showed that the higher resistance to surface penetration is achievable when using Wire A. Although there is a significant influence of the heat input on the level of hardness in both cored wires, Wire A is less prone to parameter adjustment (F_calc_ = 6.18, F_crit_ = 2.70) than the coatings manufactured with Wire B (F_calc_ = 20.33, F_crit_ = 2.45). Before the physical experiment, the authors expected different results, since Wire B contains approx. 30% of chromium and 3.5% of carbon, making this wire basically the high-chromium cast iron, which is generally considered as a hard material. However, the presence of chromium carbides can result in relatively soft, austenitic matrix of the applied alloy.

It is possible that the optimal parameters (concerning hardness of Wire B) are outside the studied scope; however, in the case of this material, wire feeds higher than 10 m/min might require a power source capable of supplying over 700 A of current, which is uncanny. Additionally, big spatter was observed in the case of every run executed with Wire B, as is presented in Figure 12.

In the chosen range of parameters, wire A exhibits the desired hardness properties. The authors also claim that within the chosen range of parameters an optimal spot has been found. The L2A sample not only has the highest mean hardness, but also offers a stable bead geometry, which is crucial for the robotized GMAW-coating-process strategy planning. The heat input value in this case was approx. 0.35 kJ/mm.

## 5. Conclusions

The authors have drawn several conclusions from the current study. The first one being that the high chromium content (~30%) in Wire B can decrease the resultant hardness value. The hypothesis for the future study is that even though the hardfaced coating contains the chromium carbides (which exhibit high hardness), the overall hardenability of the alloy is lower, whereas the lower content of chromium in Wire A enables the alloy to grow a hard, martensitic structure. Additionally, in Wire B, it was observed, that the hardness decreases in an inversely proportional manner to the heat input.

The unexpected and non-trivial conclusion from this study is that even if the weld bead geometry is affected by the heat input, higher chromium content helps to stabilize its cross-sectional shape.

The differences in the chemical composition of the studied chromium-rich cored wires changes the coating properties significantly. Considering high hardness, wire A tends to give more stable results and higher Vickers hardness values; however, the geometry of the bead is more dependent on the heat input magnitude. The smaller the heat input (or linear energy), the bigger the surface tension on the solidifying welding pool. Low values of heat result in cross-sectional geometry resembling half a circle, while coatings applied with higher energy exhibit a more elliptic sector shape. In the multi-pass WAAM technique, as long as the depth of penetration is plausible, one should aim at a half-circular shape, for it is easier to plan the toolpath of stacked and overlapping beads, maintaining the low porosity of the manufactured part. One way of achieving that shape is to find the “sweet-spot” for the particular material, which has been proven in this work. This research should be continued in order to find out about the geometry and hardness of the wires in question when applying them layer by layer.

## Figures and Tables

**Figure 1 materials-15-06019-f001:**
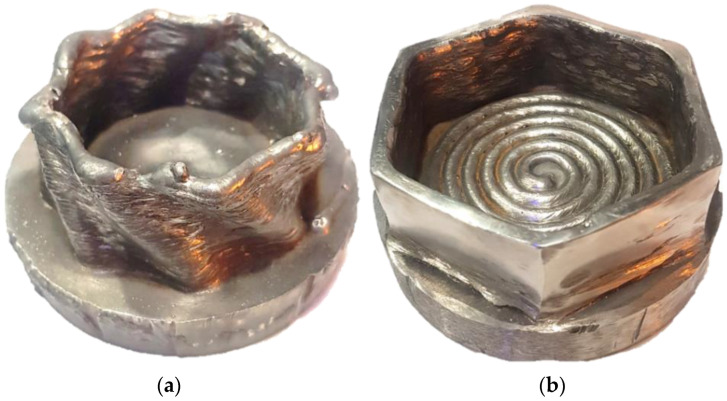
Picture of examples of WAAM-manufactured parts, “printed” by the authors. (**a**) depicts an unmachined parts, (**b**) shows an externally machined part.

**Figure 2 materials-15-06019-f002:**
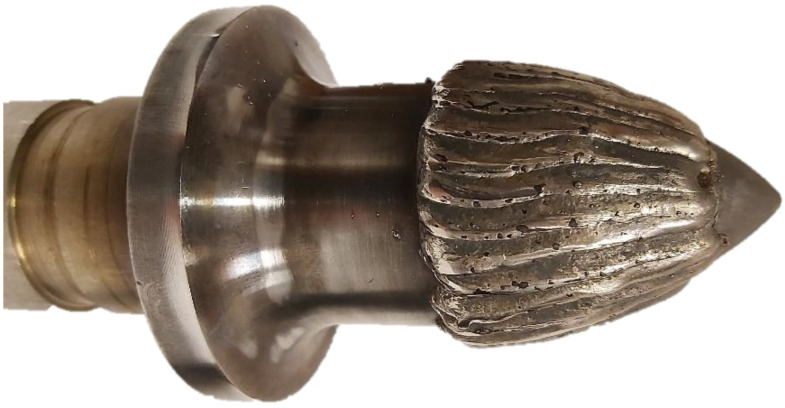
Hard coating applied with robotized GMAW process on the working surface of conical pick.

**Figure 3 materials-15-06019-f003:**
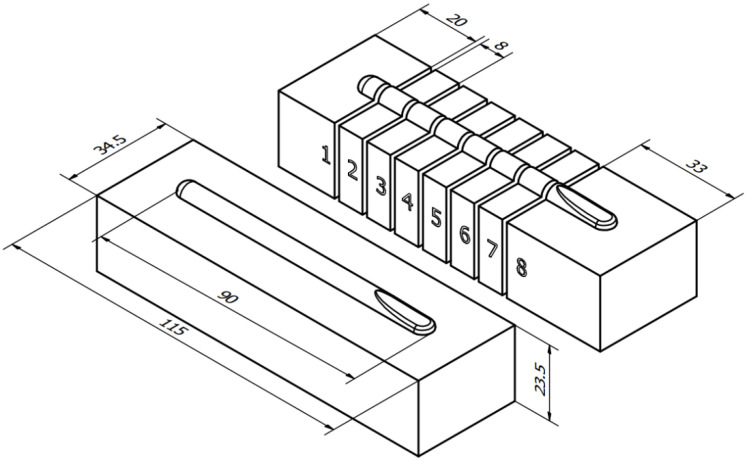
Specimen preparation scheme. The numbers from 1 to 8 indicate the separated slices of the hardfaced bar, used afterwards to measure the geometry and hardness.

**Figure 4 materials-15-06019-f004:**
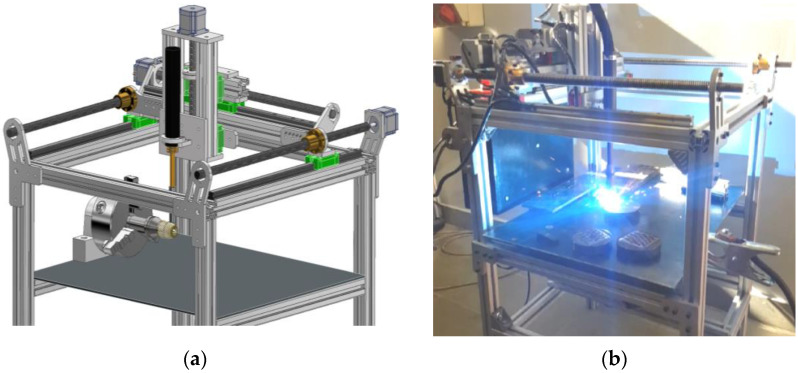
View of the CNC Hardfacing Machine, codenamed NaPawlik v2.0. (**a**) depicts the CAD model, (**b**) shows the device in action.

**Figure 5 materials-15-06019-f005:**
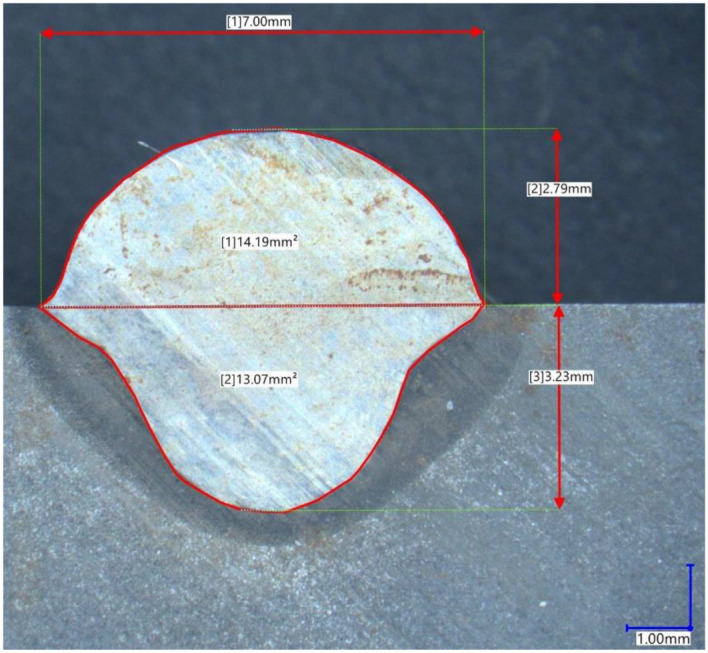
A microscopic view on the cross-sectional geometry of the weld bead.

**Figure 6 materials-15-06019-f006:**
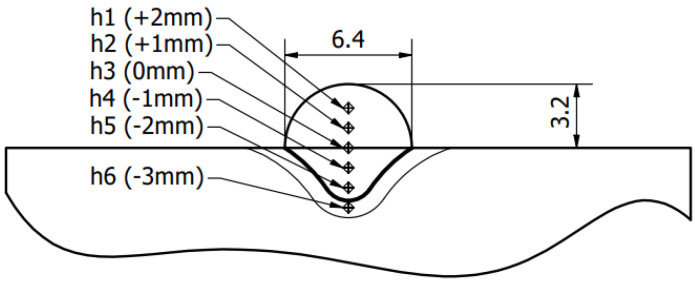
Diagram of hardness measurement.

**Figure 7 materials-15-06019-f007:**
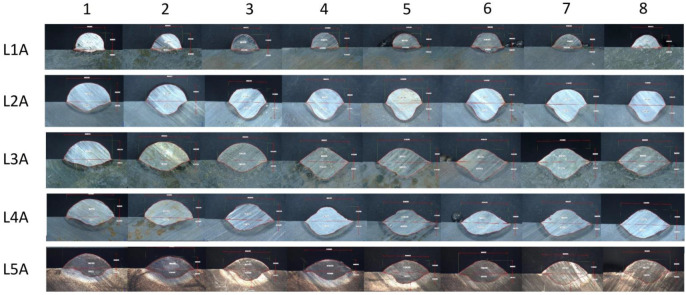
A collage of microscopic images collected for stringer bead of Wire A. The top row depicts lack of proper fusion, setting the lower limit for energy value for this material. 1–8 refer to the sample-cutting scheme, shown on Figure 3, L1A–L5A refer to Table 3.

**Figure 8 materials-15-06019-f008:**
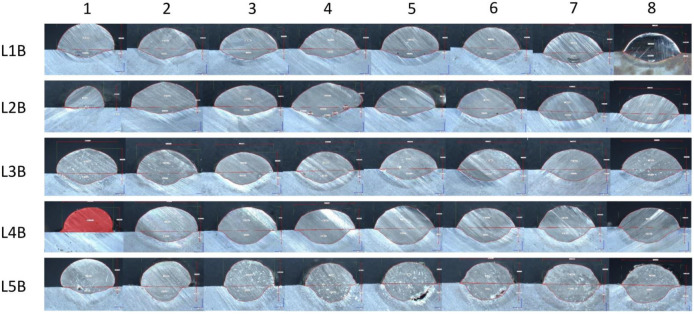
A collage of microscopic images collected for the stringer bead of Wire B. The bead fragment highlighted with red color marks the bead with welding defect (L4B, 1), namely incomplete fusion. 1–8 refer to the sample-cutting scheme, shown on Figure 3, L1B–L5B refer to Table 3.

**Figure 9 materials-15-06019-f009:**
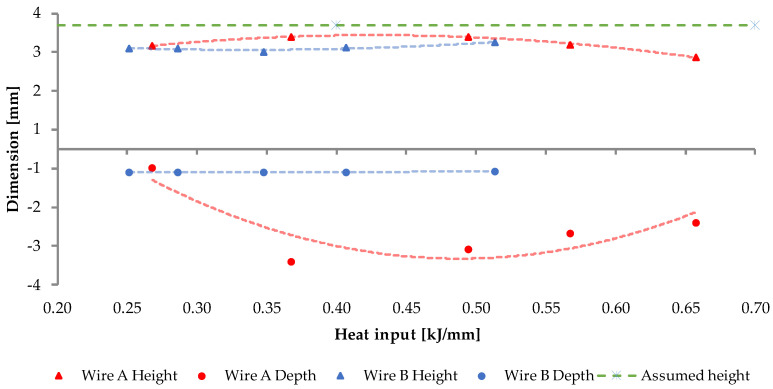
Graph representing the dependence of the weld bead height and penetration (marked as “depth”) on the heat input level. The red and blue dashed lines express the second order interpolation, while the green dashed line marks the assumed, target value for the reference.

**Figure 10 materials-15-06019-f010:**
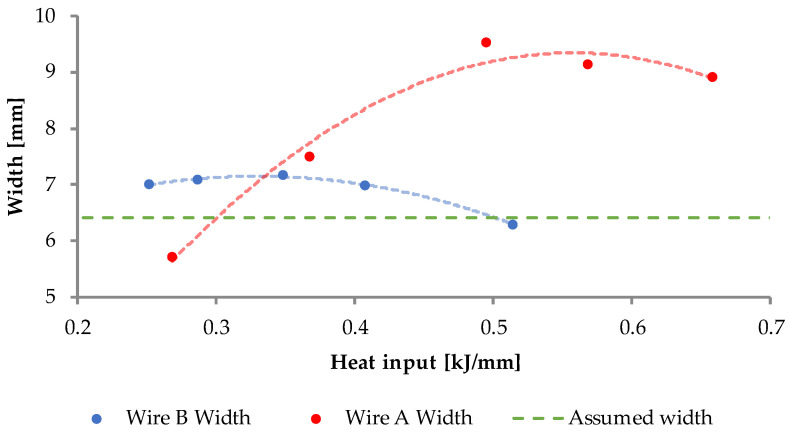
Graph representing the dependence of the weld bead width on the heat input level. The red and blue dashed lines express the second order interpolation, while the green dashed line marks the assumed, target value for the reference.

**Figure 11 materials-15-06019-f011:**
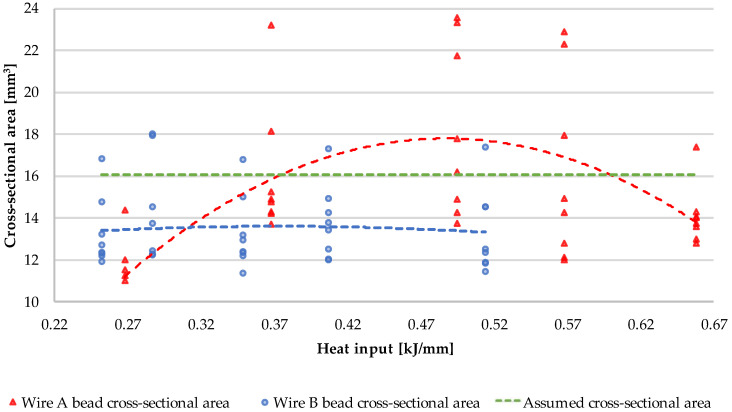
Graph representing the dependence of the cross-sectional bead area on the heat input. The red and blue dashed lines express the second order interpolation, while the green dashed line marks the assumed, target value for the reference.

**Figure 12 materials-15-06019-f012:**
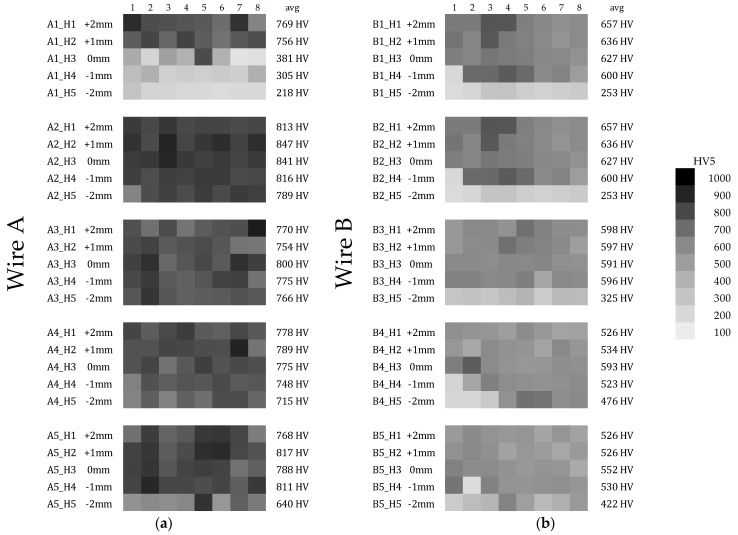
Heatmap of HV5 hardness values for: (**a**) Wire A, (**b**) Wire B. The top row A1 of (**a**) represents incomplete fusion, as there is sharp hardness difference on the base material level. The columns 1–8 correspond with the cutting scheme, depicted in Figure 3. The second level A2 of hardness values represent the “sweet-spot”. The (**b**) subfigure shows that Wire B is observably less hard, than Wire A.

**Figure 13 materials-15-06019-f013:**
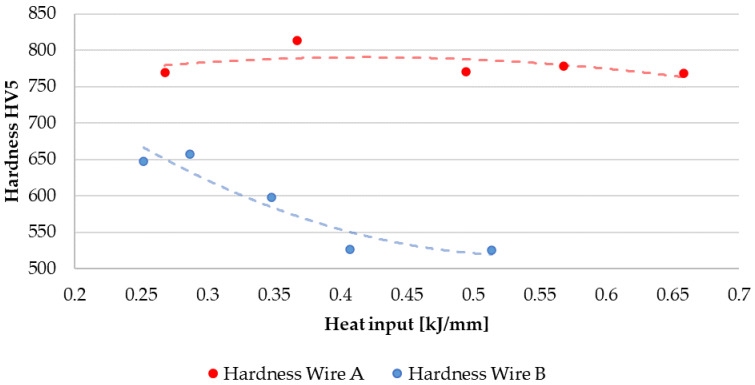
Graph of mean hardness values for Wire A and Wire B for given heat input levels. While Wire A exhibits relatively stable hardness value, the hardness of Wire B is vulnerable against the heat input level. The red and blue dashed lines express the second order interpolation.

**Table 1 materials-15-06019-t001:** Chemical composition of the C45 steel, being the base material in the current study.

Chemical Composition	wt.%
Carbon	0.42–0.5
Manganese	0.5–0.8
Silicon	0.1–0.4
Phosphorus	max. 0.04
Sulfur	max. 0.04
Chromium	max. 0.3
Nickel	max. 0.3
Molybdenum	max. 0.1
Copper	max. 0.3
Fe	bal.

**Table 2 materials-15-06019-t002:** Chemical composition of the cored wires, used in this experiment [45,46].

Chemical Component	wt.%	
	ENDOTEC DO*351 Wire A	ENDOTEC DO*332 Wire B
Carbon	0.59	3.5
Silicon	2.86	1.2
Manganese	0.56	0.6
Chromium	9.42	30.0
Molybdenum	0.01	3.8
Iron	Bal (85.56)	Bal (60.9)

**Table 3 materials-15-06019-t003:** Values for wire feed and welding head velocity selected for both kinds of materials.

Level	Wire Feed [mm/min.]	Head Velocity [mm/min.]
L1	10,000	1250
L2	8000	1000
L3	6000	750
L4	4000	500
L5	2000	250

**Table 4 materials-15-06019-t004:** Values kept constant during all experimental runs.

Material	Voltage Set [V]	Free Stick-Out [mm]	Gas Flow [L/min]	Gas Mixture
Wire A	22.1	15	12	Ar 82% + 18% CO_2_
Wire B	22.1	15	12	Ar 82% + 18% CO_2_

**Table 5 materials-15-06019-t005:** The chemical composition of the investigated cored wires (optical emission spectroscopy.

Wire Type		Chemical Composition, wt.%			
		Si	Mn	Cr	Mo
	random measurement point 1	1.74	0.84	5.57	-
Wire A	random measurement point 2	1.88	0.72	5.52	-
	random measurement point 3	1.81	1.01	6.14	-
	random measurement point 1	0.86	-	31.55	3.72
Wire B	random measurement point 2	0.79	-	38.63	3.63
	random measurement point 3	1.04	-	22.48	2.23
	Wire A mean	1.81	0.87	5.74	-
	Wire B mean	0.87	-	30.87	3.19

**Table 6 materials-15-06019-t006:** Aggregated results from ANOVA testing.

		*SS*	*df*	*MS*	*F*	*p*-Value	*F Crit*
Width A	between	78.60352	4	19.651	204.7256	9.37 × 10^−24^	2.641465
	within	3.359525	35	0.096			
Width B	between	4.99009	4	1.248	11.04536	6.72 × 10^−6^	2.641465
	within	3.953088	35	0.113			
Height A	between	1.572315	4	0.393	3.526658	0.0161	2.641465
	within	3.901075	35	0.111			
Height B	between	0.240775	4	0.0601	0.817037	0.5230	2.641465
	within	2.578563	35	0.0737			
Penetration A	between	27.81013	4	6.953	31.7829	3.28 × 10^−11^	2.641465
	within	7.656275	35	0.219			
Penetration B	between	3.10161	4	0.775	3.192195	0.0246	2.641465
	within	8.5017	35	0.243			
Hardness A	between	50,952.33	3	16,984.11	6.186054	0.0007	2.703594
	within	252,590.5	92	2745.548			
Hardness B	between	241,443.1	4	60,360.78	20.33036	1.09 × 10^−12^	2.450571
	within	341,434.6	115	2968.997			

## Data Availability

The data presented in this study are available on request from the corresponding author.

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
