# Peer review of "On the Influence of Linear Energy/Heat Input Coefficient on Hardness and Weld Bead Geometry in Chromium-Rich Stringer GMAW Coatings"

_materials, 2022, doi:10.3390/ma15176019_

Round 1

Reviewer 1 Report

While this paper has an interesting topic, it cannot be accepted in its current form due to several fundamental issues regarding explanation of the experimental methods and presentation of the results. Please address the following before resubmitting:

1)     The first four sentences of the abstract (almost half) of the abstract is spent on background information and reads more like an introduction. Maybe spend only one (at most two) sentences on background and use the rest of the abstract to summarize your results and conclusions.

2)     First sentence of the introduction: specify what industry or field you are referring to.

3)     Give a definition of hardfacing in your introduction.

4)     Speak plainly about what the welding efficiency coefficient is and how it is calculated. What is the range of values (if applicable)? What is the significance of this value?

5)     In sections 2.2 and 2.3, you go back and explain materials you introduced in section 2.1. Please combine these sections for clarity.

6)     What is the basis for your assumptions in section 2.5? Also you need to show the steps you took to get from equation 1 to equation 2. As it stands, it is too difficult to follow.

7)     In your experimental design, your wire feed to head velocity is not 1:8 as you claim. Additionally choose only one unit for your head velocity there is no need for two units to report the same value (the conversion was done incorrectly anyway).

8)     The text in Figure 5 is too small and the yellow lines are too thin. Be considerate of your readers that may have vision problems.

9)     Give units of measurement for Tables 1,2 and 5. Is it wt% or at%?

10)  What is the progression we are supposed to observe in each row in figures 7 and 8? If this is not a progression, choose a representative image. If it is a progression, state what the progression we are observing in the image description. Lastly, if you are going to include all 40 images in each figure don’t bother including text; we can’t read it anyway.

11)  In figure 11, make your markers bigger so it is easier to see. Also what are the red and blue dashed lines?

a.      Side note: consider using a different line style between series if it is not directly tied to the data

12)  Show in the weld bead pictures where the hardness values were measured, the resolution of the hardness map is too low for the reader to see anything.

13)  Please state what are the red and blue dashed lines in in figure 13.

14)  Your discussion section reads more like a conclusions section than a discussion. Please name your section accordingly or write a separate section for conclusions.

15)  Please provide references to any equations that are not your own or were derived based on someone else’s work (including your own).

Author Response

Dear Reviewer,

Thank you very much for taking the time to read our manuscript thoroughly and make recommendations for its correction and improvement. We have read the comments carefully and have responded to all your comments.

Remark 1

The first four sentences of the abstract (almost half) of the abstract is spent on background information and reads more like an introduction. Maybe spend only one (at most two) sentences on background and use the rest of the abstract to summarize your results and conclusions.

Response: Thank you for this remark. We managed to rearrange the information according to your guidance.

Remark 2

First sentence of the introduction: specify what industry or field you are referring to.
Response: We referred to the published source #1, which stated, that the wear generates costs of approx. 2.5 bln of Canadian dollars annually – thus we believe that the claim “billions of dollars a year” is a fair one.

Remark 3

Give a definition of hardfacing in your introduction.

Response: The general definition is now supplied.

Remark 4

Speak plainly about what the welding efficiency coefficient is and how it is calculated. What is the range of values (if applicable)? What is the significance of this value?

Response: Thank you for the remark, we missed the word in “welding method thermal efficiency”. This coefficient is applied according to European standards, it modifies the “linear energy” value appropriately to the utilized welding method. It ranges from n=1 (for SAW welding), 0.8 (for SMAW, GMAW and FCAW) and 0.6 for GTAW and PAW. It basically tells the user how much energy of the power source (welding machine) is transferred via electric arc to the welding puddle.

Remark 5

In sections 2.2 and 2.3, you go back and explain materials you introduced in section 2.1. Please combine these sections for clarity.

Response: Thank you, we have merged the base material part. We have decided to write about filler materials in separate subsection.

Remark 6

What is the basis for your assumptions in section 2.5? Also you need to show the steps you took to get from equation 1 to equation 2. As it stands, it is too difficult to follow.

Response: Again, thank you for the remark, now the stages of formula development is supplied.

Remark 7

In your experimental design, your wire feed to head velocity is not 1:8 as you claim. Additionally choose only one unit for your head velocity there is no need for two units to report the same value (the conversion was done incorrectly anyway).

Response: Thank you for pointing that out, it is corrected. The ratio in the text is now 1:8, as assumed for the experiment. 1250 mm/s was indeed an unachievable velocity.

Remark 8

The text in Figure 5 is too small and the yellow lines are too thin. Be considerate of your readers that may have vision problems.

Response: Thank you for the remark, we substituted the image.

Remark 9

Give units of measurement for Tables 1,2 and 5. Is it wt% or at%?

Response: it is expressed in weight percent, we added that to the table description.

Remark 10

What is the progression we are supposed to observe in each row in figures 7 and 8? If this is not a progression, choose a representative image. If it is a progression, state what the progression we are observing in the image description. Lastly, if you are going to include all 40 images in each figure don’t bother including text; we can’t read it anyway.

Response: All of the images have the role of visualization of the geometry type. Attaching all of the images would unnecessarily lengthen this paper – all of the physical, measured dimensions are supplied in the Appendix A and Appendix B. We corrected it by adding appropriate information in the text.

We claim, that this “collage” is the most important information for our future study, thus we would appreciate if you allow to publish it that way.

Remark 11

In figure 11, make your markers bigger so it is easier to see. Also what are the red and blue dashed lines?

  1. Side note: consider using a different line style between series if it is not directly tied to the data

Response: Thank you for the remark, we corrected the image.

Remark 12

Show in the weld bead pictures where the hardness values were measured, the resolution of the hardness map is too low for the reader to see anything.

Response: We believe, that Figure 6 explains the location of measurement points.

Remark 13

Please state what are the red and blue dashed lines in in figure 13.

Response: Those represent projected trend lines of the collected dataset. We added appropriate description in the captions.

Remark 14

Your discussion section reads more like a conclusions section than a discussion. Please name your section accordingly or write a separate section for conclusions.

Response: Thank you for the remark. We have rearranged the article, add some new conclusions to the Discussion section as well as added the Conclusion.

Remark 15

Please provide references to any equations that are not your own or were derived based on someone else’s work (including your own).

Response: The only formula, that was derived from somewhere else was the formula for the heat input, the appropriate source is now cited. Other formulas are basic conversions, derived from the welding efficiency expressed in kg/h.

Reviewer 2 Report

The manuscript presents an interesting study about the influence of linear energy/heat input coefficient on hardness in chromium GMAW coatings. The paper needs major revisions before it is processed further, some comments follow:

Abstract:

The abstract must be improved. Please highlight the novelty of the study. Also, when it is the first time when appears in the text don’t use an abbreviation, use the long form (SAW, GMAW, TIG). Also, please write the information from the last paragraph of the introduction, shortly, in the abstract.

Introduction:

The introduction section must be improved.

Also, multiple citations have been introduced in bulk form "[5-8]", "[12-15]", "[16-18]" , "[24-27]", "[30-33]", "[34-36]"   and not distributed in the text in accordance with the affirmations that must be supported. Please introduce citations in a specific position to ensure clear correspondence between the affirmations from the introduction section and the previous publication. Moreover, to avoid this type of citing, please cite review type of studies.

Results

Subsection 3.1. In order for the results to be more realistic, the chemical composition measurement must be done at different points on the surface of the material. Please write if the authors measure in this way or if not please redo the measurements.

Discussion

The discussions are poor. Please improve by comparing these results with other studies.

Conclusion

The conclusion section is missing.

Author Response

Dear Reviewer,

Thank you very much for taking the time to read our manuscript thoroughly and make recommendations for its correction and improvement. We have read the comments carefully and have responded to all your comments.

Abstract:

Please highlight the novelty of the study.

Response: Thank you for the remark. The novelty of the article was added to Introduction as well as to the Conclusion section

Also, when it is the first time when appears in the text don’t use an abbreviation, use the long form (SAW, GMAW, TIG).

Response: Thank you for the remark. It was corrected

Also, please write the information from the last paragraph of the introduction, shortly, in the abstract.

Response: Thank you for the remark. I was added to the article

Introduction:

Also, multiple citations have been introduced in bulk form "[5-8]", "[12-15]", "[16-18]" , "[24-27]", "[30-33]", "[34-36]" and not distributed in the text in accordance with the affirmations that must be supported. Please introduce citations in a specific position to ensure clear correspondence between the affirmations from the introduction section and the previous publication. Moreover, to avoid this type of citing, please cite review type of studies.

Response: Thank you for the remark. The citations are now split, however we previously introduced some of them in bulk, since - for instance - the sources regarding Fronius CMT technology were included because even if it is theoretically still GMAW technology it relies on thoroughly different approach towards the weld bead formation. Nevertheless, we corrected the paper according to Your remark.

Results

Subsection 3.1. In order for the results to be more realistic, the chemical composition measurement must be done at different points on the surface of the material. Please write if the authors measure in this way or if not please redo the measurements.

Response: Thanks again for the remark, we forgot to include information about random-point measurement. In our original text it was written as “Wire A sample 1”, where we meant “measurement point 1”. It is corrected now.

Discussion

The discussions are poor. Please improve by comparing these results with other studies.

Response: Thank you for the remark. The Discussion section has been improved significantly according to your remark

Conclusion

The conclusion section is missing.

Response: Thank you for the remark. The conclusion section has been added

Round 2

Reviewer 1 Report

The paper has improved significantly from the previous version. However, a few of the concerns from the previous version remain:

Remark 2 original text: “First sentence of the introduction: specify what industry or field you are referring to.

·        Your sentence reads “The advancement in the industry”. The remark from the original report is asking what is “the industry” you are referring to. Is it the aerospace industry? Automotive industry? A reference is fine, but your readers should not have to read the reference to understand the context of your first sentence. If this applies to multiple industries, rewrite it to say which industries you are referring to. For example, you can say “Machine wear inhibits the advancement in several industries such as the automotive and aerospace industries”

Remark 10 original text: “What is the progression we are supposed to observe in each row in figures 7 and 8? If this is not a progression, choose a representative image. If it is a progression, state what the progression we are observing in the image description. Lastly, if you are going to include all 40 images in each figure don’t bother including text; we can’t read it anyway.”

·        The original comment was not to separate all 40 images, it was to clarify what the reader is looking at. If each image was taken at the 8 slices of the weld bead you described in you Section 2, then you need to state that! Your readers will not immediately remember the details of Figure 3 without a reminder. Please add numbers (1-8) above each column of images to designate which cut corresponds to each image. Also stating somewhere that these were the slices indicated in Figure 3 would be helpful.

Author Response

Dear Reviewer,

Thank you very much for the help with paper enhancement and very fast revision.

Remark 2 original text: “First sentence of the introduction: specify what industry or field you are referring to.”

 Your sentence reads “The advancement in the industry”. The remark from the original report is asking what is “the industry” you are referring to. Is it the aerospace industry? Automotive industry? A reference is fine, but your readers should not have to read the reference to understand the context of your first sentence. If this applies to multiple industries, rewrite it to say which industries you are referring to. For example, you can say “Machine wear inhibits the advancement in several industries such as the automotive and aerospace industries”

Response: Thank you for the remark, I must have previously understood it in a wrong manner. We wanted to write a sentence about the “arch-reason” behind our anti-wear efforts. In our opinion there is hardly any branch that does not suffer from different wear mechanisms. We supplemented the original text.

Remark 10 original text: “What is the progression we are supposed to observe in each row in figures 7 and 8? If this is not a progression, choose a representative image. If it is a progression, state what the progression we are observing in the image description. Lastly, if you are going to include all 40 images in each figure don’t bother including text; we can’t read it anyway.”

        The original comment was not to separate all 40 images, it was to clarify what the reader is looking at. If each image was taken at the 8 slices of the weld bead you described in you Section 2, then you need to state that! Your readers will not immediately remember the details of Figure 3 without a reminder. Please add numbers (1-8) above each column of images to designate which cut corresponds to each image. Also stating somewhere that these were the slices indicated in Figure 3 would be helpful.

Response: Thank you for the remark – we believe it is now more legible for the Readers. The information is appropriately commented in the text, and also the column indicators are added.

Reviewer 2 Report

The authors addressed all my comments. The paper can be accepted as it is. 

Author Response

Dear Reviewer,

Thank you very much once again for reviewing our article and help with its enhancement!

Authors